# Nutrient Composition and Antioxidant Activity of *Cercis chinensis* Flower in Response to Different Development Stages

Hong-Yu Ren [1], Wen-Zhang Qian [1], Lu Yi [1], Yu-Lin Ye [1], Tao Gu [1], Shun Gao [1,2,*] and Guo-Xing Cao [1,2,*]

[1] Department of Forestry, Faculty of Forestry, Sichuan Agricultural University, Chengdu 611130, China; renhongyu@stu.sicau.edu.cn (H.-Y.R.); 202001247@stu.sicau.edu.cn (W.-Z.Q.); yilu@stu.sicau.edu.cn (L.Y.); yeyulin@stu.sicau.edu.cn (Y.-L.Y.); gutao1@stu.sicau.edu.cn (T.G.)

[2] National Forestry and Grassland Administration Key Laboratory of Forest Resources Conservation and Ecological Safety on the Upper Reaches of the Yangtze River, Sichuan Province Key Laboratory of Ecological Forestry Engineering on the Upper Reaches of the Yangtze River, Sichuan Agricultural University, Chengdu 611130, China

\* Correspondence: shungao@sicau.edu.cn (S.G.); cgxing@sicau.edu.cn (G.-X.C.)

**Abstract:** *Cercis chinensis* Bunge (*C. chinensis*), well known as an ornamental plant widely distributed in China, and its flowers, bark, fruit, etc., have multiple bioactivities. However, reports on the changes in mineral elements, nutrient composition and antioxidant activity in *C. chinensis* flower at different development stages are rare. In this study, the flower samples were collected every 20 days from March 2023 to May 2023. The changes in carbon (C), nitrogen (N), phosphorous (P), soluble protein (SP), amino acid (AA), non-structural carbohydrate (NSC), total phenol (TP) and total flavonoids (TF) content as well as antioxidant activity in *C. chinensis* flower at different development stages were investigated. The results suggested that C, N, and P content, the C:N:P stoichiometric ratio, NSC contents, SP, AA, TP, TF and antioxidant activity of flower showed large variations at three development stages. This study found that C and P contents showed a significant decrease with the development and opening of flowers, while N content showed an opposite trend. The soluble protein content first decreased and then increased. The amino acid content, total polyphenol content, and total flavonoid content all showed a significant downward trend, while the content of NSC increased. Both ABTS and ferric-reducing antioxidant power (FRAP) showed significant decreases at various developmental stages, but DPPH was completely opposite. The highest NSC content and DPPH activity were observed at stage III, but the highest AA, TP, TF, as well ABTS and FRAP activity wereobserved at stage I. These findings will improve understanding of the requirements and dynamic balance among C, N, and P, NSC and nutrient contents as well as antioxidant activity of *C. chinensis* flowers in response to development stages.

**Keywords:** *Cercis chinensis*; flowerdevelopment; mineral element; phenol; flavonoids; antioxidant activity





## 1. Introduction

*Cercis chinensis* (*C. chinensis*) Bunge belongs to Fabacee *Cercis*, which is native to China and widely distributed in China. It not only has high ornamental value, but also has a long history of utilization as a traditional Chinese medicine, of which various parts can be used. For example, the bark and wood of *C. chinensis* can be used to activate blood, restore menstruation, reduce swelling and detoxification, and the fruit can be used to treat cough [1,2]. In the same way, the flower can be used to treat bronchitis, intestinal worms, fungal infections, hepatitis, dysentery, liver diseases, diarrhea, Leprosy, skin diseases, wounds, tumors and bacterial infections [1,3]. The main chemical components of the flower are flavonoids, phenolic acids, toluene, lignin and polysaccharides, which show the highest antioxidant activity in terms of scavenging DPPH free radicals, ABTS free radicals and reducing iron ions [3]. In addition, studies have found that two homogeneous heteropolysaccharides, with an average molecular weight of 17,060 and 8303 Da, can be

isolated from *C. chinensis*. They are all effective flocculants that can significantly shorten activated partial thromboplastin time, prothrombin time, and thrombin time, exerting a coagulation promoting effect [4]. Many studies have shown that there is anthocyanin in *C. chinensis* flowers, so it is widely used for the extraction of natural edible red pigment [5].

Edible flowers are rich sources of phytochemistry, containing protein, amino acids, phenols and other nutrients, and have great growth potential in the food industry and medicine [6]. Edible flowers have potential additive effects in preventing chronic diseases, promoting health, and preventing food oxidation. However, as a natural antioxidant, edible flowers need to be further studied in terms of the antioxidant mechanism, anti-tumor, anti-inflammatory and anti-aging activities [7]. Flowering is one of the most important physiological processes in plants, and the changes in primary and secondary metabolite may play a regulatory role in flower development [8–10]. According to the types of flower, the flower can be consumed as a bud, whole flower and/or petals. The transition from closing bud to opening flower is one of the most active growth stages in the entire development process of plants [11]. Every flower, at each developmental stage, exhibits unique ingredients and nutritional value, which can serve as a new source in the food, cosmetics, and pharmaceutical industries [12].

The supply of sugar is all that is needed to drive flower development. The soluble sugar (SS) content also increases, before or during the rapid expansion of flowering [11,13]. Starch is the main storage carbohydrate in flowering plant, which plays an important role in pollen tube growth, ovule and fruit formation, and flower quality determination, one of the nutrients that may also be converted into SS during flower bud differentiation to meet metabolic energy needs [14,15]. Similarly, soluble protein (SP) is also one of the important energy sources of plants and the basis of morphogenesis of flower organs [16]. Amino acids (AA) also play an important role in flowering and are used to synthesize secondary metabolites, including intracellular signaling molecule, structural proteins and enzymes [9,10]. Plant secondary metabolites are essential molecules for plant growth, development, reproduction, and protection, especially phenolic compounds. They have biological effects such as free radical scavenging, antioxidant, antiviral, anti spasmodic, anti-inflammatory, and antibacterial activities, and are beneficial to human health [17]. Flavonoid have various health promoting effects and are important ingredients in many food supplements, medicines and cosmetics [18]. The various elements contents in plants are correlated, and sufficient elemental contents and relatively stable stoichiometric ratios are crucial for the healthy growth and development of plants [7,19]. Carbon (C) is the basic that constitutes cells, tissues and organs, and provides energy for various life activities of plants [20]. Nitrogen (N) and phosphorus (P) are particularly important among various nutrient elements required by plants, which are nutrient limiting indicators for biomass production, and important factors that limit physiological activities such as plant growth and reproduction [21]. They are not only components of many important organic compounds in plants, but also participate in various metabolic processes in plants.

Various cultures around the world consume flowers as food, as part of traditional cuisine or alternative medicine, because their nutritional characteristics, especially their antioxidant compound content, can play an important role in promoting health and preventing different diseases. The nutritional characteristics and antioxidant capacity of flowers vary at different stages.The purpose of this study is to investigate the changes in mineral elements and nutrient contents as well as antioxidant activity of *C. chinensis* flower at different development stages. The results will help to better understand the changes in nutrients and antioxidant activity of *C. chinensis* flower during flower development, and find the best harvest period to develop its functional ingredients in medicine and the food industry, and provide a new plant source for the edible flower market.

## 2. Materials and Methods

### 2.1. Study Site

The experiment was conducted in Sichuan Agricultural University, District Wenjiang, Chengdu, China. It has a mid-latitude inland subtropical monsoon climate with a mild climate, four distinct seasons and abundant rainfall. The annual average rainfall is 896.1 mm, and the rainy season is mainly from June to September. The annual average temperature is 16.4 °C, the annual average relative humidity is 84.0%, the annual average sunshine duration was 1104.5 h, and the annual frost-free period is 282 days.

### 2.2. Chemicals

We used the Milli-Q system (Millipore Corp., Billerica, MA, USA) to produce ultrapure water. Ethanol, Folin-Ciocalteu reagent, gallic acid, bovine serum albumin, glucose, Iron(III) chloride hexahydrate ($FeCl_3·6H_2O$), 1,1-diphenyl-2-picrylhydrazine (DPPH) and 2,2′- azido bis (3-ethyl Benzothiazole lin-6-sulfo propionic acid) (ABTS) were purchased from Sigma Aldrich (St. Louis, MO, USA). The other reagents are analytical-grade reagents purchased from China National Pharmaceutical Group Chemical Reagent Co., Ltd. (Shanghai, China).

### 2.3. Experiment Design and Plant Materials

From March 2023 to May 2023, samples every 20 days were collected from more than three trees at the campus of Sichuan Agricultural University in Wenjiang District, Chengdu, China. The Biologische Bundesantalt, Bundes-sortenamt and Chemische Industrie scales (BBCH) represent a unified coding system for describing phenologically similar growth stages in mono- and dicotyledonous plants [22]. Referring to the BBCH edited by Mishchenko and Rana et al. [23,24], the flower development of *C. chinensis* was coded. Flower development was divided into three stages: stage I—closed bud flower clustered tightly, showing some petal color at the tip of the bud end (Figure 1A); stage II—closed bud flower grown and expanded, inflorescences scattered, showing the color of petals in their entire surface (Figure 1B); stage III—completely opened flower, without symptoms of senescence (Figure 1C). The flower samples were collected from these three stages. Samples were dried at 80 °C for 48 h, and the relative water content was determined. Samples were also used for subsequent experiments. The samples are stored at −60 °C.The transverse diameter, longitudinal diameter and biomass of flower buds in three stages were measured by using the vernier scale. Each measurement consists of three technical repetitions, and then the average of the three values is calculated.

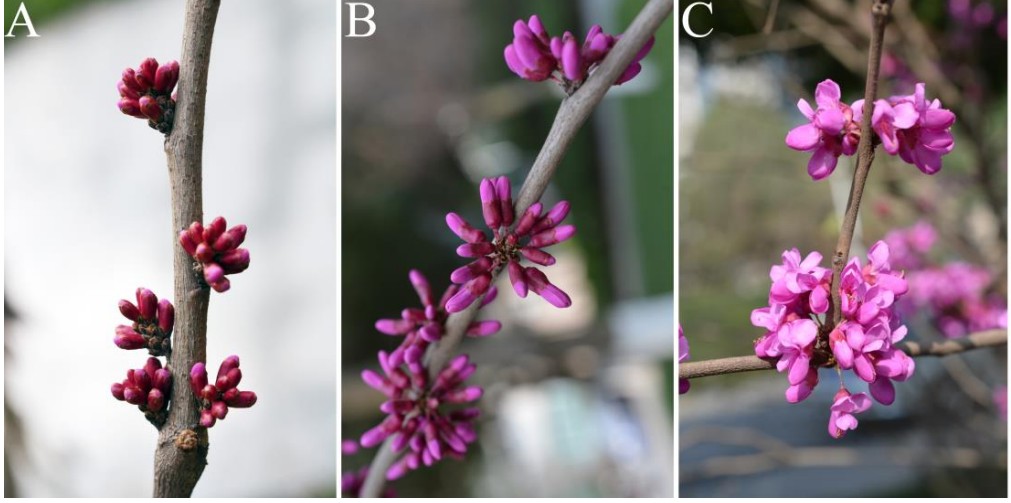

**Figure 1.** *C. chinensis* flower at different development stages. (**A**) stage I, (**B**) stage II, and (**C**) stage III.

### 2.4. Determination of C, N and P Contents

The content of total C is determined by the potassium dichromate oxidation method. The contents of N and P was determined by the sulfuric acid–perchloric acid digestion method. The total N content is determined by the Kjeldahl method. The molybdenum antimony colorimetric method was used to determine the total P content [25]. The concentration of C, N and P is expressed in g/kg dry weight (DW), and the C:N, C:P, and N:P ratios are expressed on a mass basis.

### 2.5. Determination of Protein and Amino Acid Content

Protein content was determined by the Coomassie brilliant blue method [26]. An amount of 0.5 g of sample was put into a mortar and ground into a homogenate with distilled water, centrifuged, part of the supernatant was added to Coomassie brilliant blue, left for 15 min, and then we measured the absorbance value of the sample with a spectrophotometer at 595 nm. The protein content in the sample was obtained according to the standard curve and formula (Table S1). Quantification of soluble protein was performed by following the Coomassie brilliant blue G-250 method using bovine serum albumin as a standard. The results are expressed in mg/g fresh weight (FW).

Amino acid content was determined by the ninhydrin color method [9]. An amount of 0.5 g of fresh leaf was homogenized with 5 mL 10% acetic acid, and the extraction was centrifugated at 12,000 rpm at 4 °C. A volume of 0.5 mL supernatant of per sample was placed in a test tube, and 2 mL of distilled water, 3.0 mL of ninhydrin hydrate and 0.1 mL of ascorbic acid were added. The reaction mixtures were heated in boiling water for 15 min, cooled to room temperature, and then diluted to 20 mL with 60% ethanol. Absorbance was measured at 570 nm. The results are expressed in mg/g FW (Table S1).

### 2.6. Determination of NSC Content

The NSC content was determined using the anthrone colorimetric method [27].The soluble sugar content was determined using the anthrone colorimetric method. An amount of 0.2 g of samples was mixed with 10 mL of 80% ethanol, and the mixture was incubated in a boiling water bath for 30 min, and then centrifuged at 5000 rpm for 10 min. The supernatant was collected, and this process was performed three times to ensure complete sugar extraction. These supernatants were added to achieve a 50 mL constant volume for measuring soluble sugar content. For glucose measurement, 5 mL of anthrone reagent was added to the 0.1 mL soluble sugar extraction liquid and placed in a 90 °C water bath for 15 min. For sucrose measurement, 0.1 mL of sugar extract and 0.1 mL of 7.6 mol/L KOH solution were mixed, and incubated at 100 °C for 15 min. After cooling, 5 mL of anthrone solution was added, and placed in a 90 °C water bath for 15 min, and then the absorbance at 620 nm was recorded. For fructose measurement, 0.1 mL of sugar extract and 5 mL of anthrone solution were mixed at 25 °C for 90 min. For starch measurement, add 10 mL of 30% perchloric acid to the precipitate after centrifugation of soluble sugars. Let it stand overnight, and then extract accurately in an 80 °C water bath for 10 min. After cooling, centrifuge and collect the supernatant. The measurement steps were the same as above the glucose. These reaction mixtures were measured at 620 nm, and the contents were calculated as the glucose, sucrose, fructose and starch standard curves, respectively (Table S1). The results were expressed as μg per mg of sample. The results are expressed in mg/g DW.

### 2.7. Determination of Total Phenol (TP) and Total Flavonoid (TF) Contents

The total phenol (TP) content is estimated by the Folin-Ciocalteau method [28]. The content of polyphenols was determined using the Folin phenol method. A sample of 0.2 g was taken and ground into a homogenate with 60% ethanol in a mortar. The mixture was then placed in a centrifuge tube and boiled in a water bath for 30 min. After cooling and centrifugation, the supernatants were mixed with Folin phenol reagent, 7.5% sodium carbonate solution, and distilled water. After avoiding light for 2 h at room temperature,

the absorbance value was measured at 765 nm using a spectrophotometer. The calculation result is based on the calibration curve of gallic acid and expressed as the gallic acid equivalent mg/g DW (Table S1).

The total flavonoid (TF) content was determined using Yang's method [29]. A sample of 0.3 g of ground samples was extracted three times using 10 mL of 60% ethanol at 65 °C, and the prepared extraction was centrifuged at 4000 rpm for 10 min at 4 °C. These supernatants were added to achieve a 50 mL constant volume for measuring the TF content. A volume of 1 mL of extract sample was mixed with 5 mL of 60% ethanol, and 300 μL of a 5% $NaNO_2$ solution. After 6 min, 300 μL of 10% $AlCl_3$ solution was added. After 6 min, 4 mL of 1 M NaOH and 400 μL of distilled water were added to prepare the reaction mixture. These solutions were mixed well and the absorbance was read at 510 nm. The results were calculated based on the calibration curve of rutin and expressed as the rutin equivalent mg/g DW (Table S1).

### 2.8. Assay of DPPH Radical-Scavenging Activity

The DPPH radical-scavenging rate was measured using the Gulcin method [30]. In the preparation method, 0.1 mM DPPH:2 mL of DPPH solution plus 100 μL sample solution, and 1 mL Tris-HCl (pH7.4) were mixed evenly, and stored in the dark for 30 min at room temperature. The absorbance values were recorded at 517 nm. The antioxidant capacity is expressed as the tea polyphenol equivalent (GTP mg/g DW) according to the standard curve of tea polyphenol (GTP) antioxidant activity (Table S1). The rate of DPPH radical-scavenging capability (%) = $(1 - A/A_0) \times 100\%$, where $A_0$ is the absorbance of the control and A is the absorbance of the sample extract.

### 2.9. Assay of ABTS Radical-Scavenging Activity

The ABTS radical-scavenging activity was measured using the method described by Srelatha and Padma [31]. The ABTS solution was prepared by mixing 7 mM of ABTS with 2.45 mM of potassium persulfate, and allowed to react at room temperature in the dark for 16 h. The stock solution was diluted with ethanol to an absorbance of $0.70 \pm 0.02$ at 734 nm. TP extracts (200 μL) were allowed to react with 800 μL of ABTS for 6 min, and then the absorbance was measured at 734 nm. The deionized water was used as the control. The antioxidant capacity is expressed as the tea polyphenol equivalent (GTP mg/g DW) according to the standard curve of tea polyphenol (GTP) antioxidant activity (Table S1). The rate ABTS free radical-scavenging activity (%) = $(1 - A/A_0) \times 100\%$, where $A_0$ is the absorbance of the control and A is the absorbance of the sample extract.

### 2.10. Assay of Ferric-Reducing Antioxidant Power (FRAP)

The FRAP was measured using El Karkouri's method [32]. The reaction solutions included 2 mL of sample, 2 mL of 1% potassium ferrocyanide, and 2 mL of 0.2 M phosphate buffer (pH 6.6), and then were mixed evenly, and then place the centrifuge tube in warm water at 50 °C for 20 min. After cooling, the 2 mL of 10% TCA, deionized water and 0.3 mL of 0.1% iron chloride were added, and mixed evenlyfor reaction for 10 min. The absorbance values were recorded at 700 nm. The antioxidant capacity was expressed as the tea polyphenol equivalent (GTP mg/g DW) according to the standard curve of tea polyphenol (GTP) antioxidant activity (Table S1).

### 2.11. Statistical Analysis

Experiments were carried out in a randomized way with three replicates. The data were analyzed using one-way analysis of variance (one-way ANOVA), and expressed as the means $\pm$ SD. One-way analysis of variance (ANOVA) was performed using the Waller–Duncan multi-interval test using SPSS26.0 software (IBM® Corporation, USA). Moreover, the minimum significant difference (LSD) in the ANOVA test was used to analyze the significant difference. Statistical significance was set at a 95% confidence level ($p < 0.05$).

## 3. Results

### 3.1. Changes in Morphological Features and Biomass, and Water Content

As shown in Figure 2, the transverse diameter of *C. chinensis* flowers significantly increased at each stage from stage I to stage III, ranging from $3.22 \pm 0.32$ mm to $13.99 \pm 0.30$ mm. The longitudinal diameter showed the same trend, but there was no significant change between stage II and stage III, ranging from $8.06 \pm 0.50$ mm to $13.19 \pm 0.67$ mm (Figure 2A,B). The biomass of *C. chinensis* flowers also significantly increased with each stage from stage I to stage III, ranging from $27.7 \pm 1.53$ mg to $68.00 \pm 0.40$ mg (Figure 2C). The water content of *C. chinensis* flowers showed a significant increase in each period from stage I to stage III (Figure 2D). There are obvious changes in the morphology of *C. chinensis* flowers during three developmental stages. At stage I, the petals are arranged tightly, forming female stamens. At stage II, the length and width of petals increase, the arrangement becomes loose, and the pistils and stamens are in the stage of development and elongation. At stage III, the *C. chinensis* flowers fully open and the pistils and stamens fully develop. In the three developmental stages of *C. chinensis* flowers, from one stage to the next, there is an increase in biomass and water content.

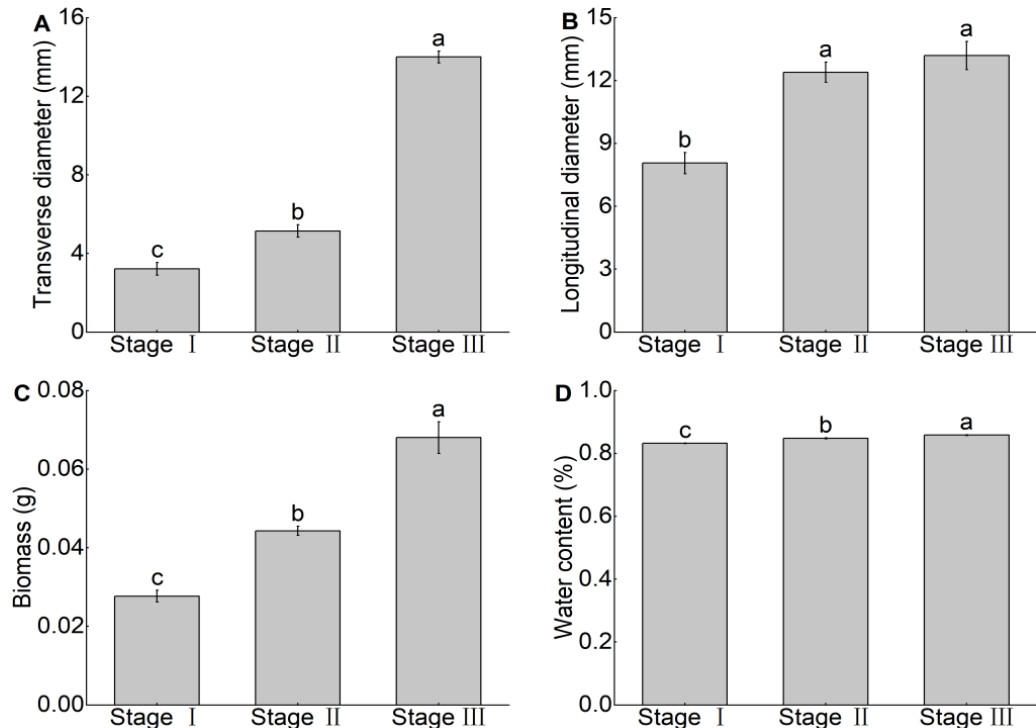

**Figure 2.** Changes in transverse diameter, longitudinal diameter, biomass and water content of *C. chinensis* flower at different development stages. (**A**) Transverse diameter, (**B**) longitudinal diameter, (**C**) biomass, and (**D**) water content. Data represent the mean $\pm$ SEM, $n = 3$. Different lowercase letters (a–c) represent significant differences among different development stages ($p < 0.05$).

### 3.2. Changes in C, N, and P Content and the C:N:P Stoichiometric Ratio

As shown in Figure 3A, the C content in *C. chinensis* flowers shows a significant downward trend with flower development and opening ($467.46 \pm 7.43$ g/kg to $410.17 \pm 9.95$ g/kg), and shows significant differences in all three stages. Similarly, the P content also showed a significant decrease with the development and opening of flowers ($1.45 \pm 0.11$ g/kg to $0.91 \pm 0.02$ g/kg), and the P content at stage I was significantly higher than that at stages II and III (Figure 3C). In Figure 3B, the range of N content in the three developmental stages of *C. chinensis* flowers follows the floral development stage, but the fluctuation is relatively gentle and the difference is not significant ($24.29 \pm 0.53$ g/kg to $26.26 \pm 1.37$ g/kg). The C:N ratio showed a continuous decreasing trend during the three developmental

stages of *C. chinensis* flowers, but the difference between stages I and II was not significant, with 19.25 ± 0.55 and 17.83 ± 0.14, respectively. However, it significantly decreased to 15.66 ± 1.12 during stage III (Figure 3D). However, the C:P and C:N ratios are indeed the opposite. The C:Pratio showed a continuous upward trend during the three developmental stages. The ratio significantly increased from 322.67 ± 23.36 to 451.09 ± 4.12 between stages I and II, and reached its highest point at 452.54 ± 16.76 during stage III. However, the upward trend between stages II and III was relatively small (Figure 3E). The N:P ratio showed a very significant stepwise increase in all three stages, with values of 16.79 ± 1.63, 25.30 ± 0.19, and 28.96 ± 1.12, respectively (Figure 3F).

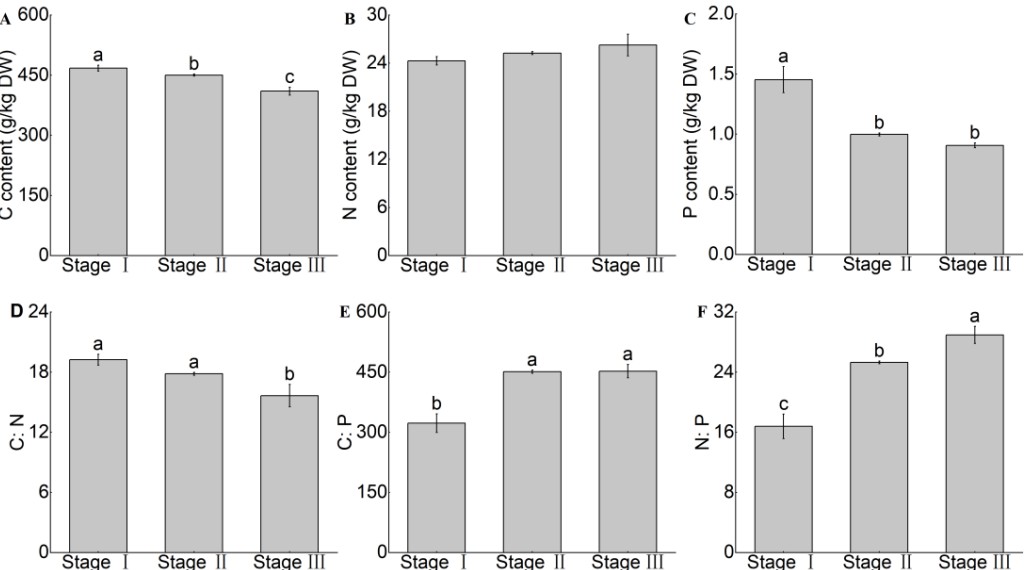

**Figure 3.** Changes in C, N, and P contents and C:N:P stoichiometry of *C. chinensis* flower at different development stages. (**A**) C contents, (**B**) N contents, (**C**) P contents, (**D**) the C:N ratio, (**E**) the C:P ratio, and (**F**) the N:P ratio. Data represent the mean ± SEM, $n$ = 3. Different lowercase letters (a–c) represent significant differences among different development stages ($p < 0.05$).

### 3.3. Changes in NSC Content

As shown in Figure 4, the content of NSC continued to increase from stage I to stage III, and there was a significant increase at each stage, ranging from 106.74 ± 7.55 mg/g to 169.35 ± 14.77 mg/g (Figure 4A). The soluble sugar showed a decreasing trend, reaching 48.72 ± 2.55 mg/g at stage I, 45.51 ± 1.46 mg/g at stage II, and 63.60 ± 5.59 mg/g at stage III (Figure 4B). Starch was the lowest at stage I (58.01 ± 7.43 mg/g), and significantly increased at stage II (96.01 ± 7.94 mg/g), and reached its highest level at stage III (105.75 ± 13.00 mg/g) (Figure 4C).

### 3.4. Changes in Soluble Protein, Amino acid, TP and TF Content

As shown in Figure 5A, the soluble protein content was 0.68 ± 0.06 mg/g at stage I, decreased to 0.67 ± 0.11 mg/g at stage II, and then reached its highest level of 0.78 ± 0.01 mg/g at stage III. The soluble protein content showed at first decrease and then increase during the three developmental stages of *C. chinensis*, but there was no significant change between them. In Figure 5B, the decrease in amino acids during each of the three periods is very significant, ranging from 7.21 ± 0.35 μmol/g to 3.19 ± 0.08 μmol/g. According to Figure 5C, it can be seen that the TP content shows a significant decrease trend with the development stage of *C. chinensis* flowers, and shows significant differences in all three stages (55.75 ± 5.79 mg/g to 10.54 ± 3.15 mg/g). The highest TF content at stage I was 9.51 ± 1.00 mg/g, significantly higher than those of 3.73 ± 0.20 mg/g and 2.33 ± 0.64 mg/g levels at stages II and III. Similar to the content of polyphenols, the flavonoid content decreased from the tight bud stage to the fully open stage (Figure 5D).

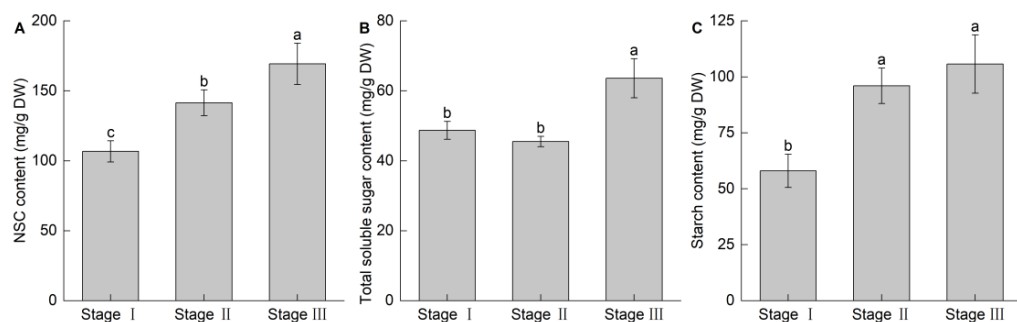

**Figure 4.** Changes in NSC, total soluble sugar, and starch contents of *C. chinensis* flower at different development stages. (**A**) NSC contents, (**B**) Total soluble sugar contents, (**C**) Starch contents. Data represent the mean ± SEM, *n* = 3. Different lowercase letters (a–c) represent significant differences among different development stages (*p* < 0.05).

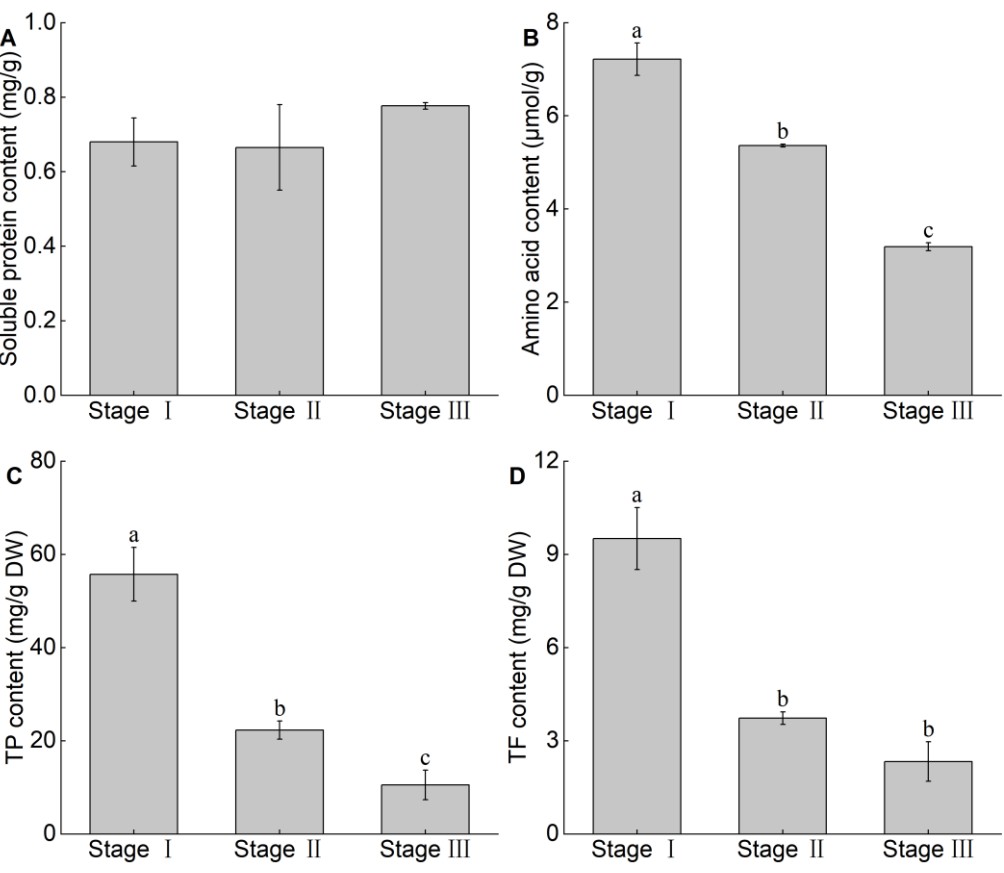

**Figure 5.** Changes in soluble protein, amino acid, TP and TF contents of *C. chinensis* flower at different development stages. (**A**) Soluble protein contents, (**B**) Amino acid contents, (**C**) TP contents, (**D**) TF contents. Data represent the mean ± SEM, *n* = 3. Different lowercase letters (a–c) represent significant differences among different development stages (*p* < 0.05).

### 3.5. Changes in Antioxidant Activity

Both ABTS cation radical and FRAP showed significant decreases at various developmental stages of *C. chinensis*, ranging from 18.81 ± 0.97 mg/g to 4.97 ± 1.31 mg/g and 4.67 ± 0.11 mg/g to 1.80 ± 0.40 mg/g, respectively (Figure 6A,C). The difference is that at Figure 6B, the DPPH radical showed a significant increase at stages I to III (9.29 ± 0.03 mg/g to 9.38 ± 0.01 mg/g). Both ABTS and FRAP show significant decreases at various developmental stages of *C. chinensis*. However, DPPH is different, showing a significant increase from stage I to stage III.

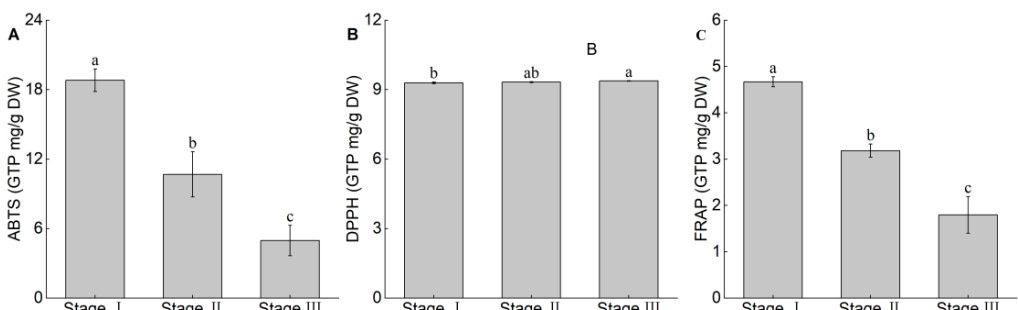

**Figure 6.** Changes in antioxidant activity of *C. chinensis* flower extraction at different development stages. (**A**) ABTS radical-scavenging activity, (**B**) DPPH radical-scavenging activity, (**C**) FRAP. Data represent the mean ± SEM, *n* = 3. Different lowercase letters (a–c) represent significant differences among different development stages (*p* < 0.05).

### 3.6. Correlation Analysis

As shown in Figure 7, at three stages of flower development, NSC, C, N, P, the C:N ratio, the N:P ratio, the C:P ratio, total polyphenol, total flavone, amino acid, ABTS, DPPH and FRAP are closely related and have significant effects on each other (*p* < 0.05, *p* < 0.01), but there is no significant relationship between the content of soluble protein and other nutrients, mineral elements and antioxidant activity (*p* > 0.05). C, P content had a significant negative correlation with NSC and DPPH (*p* < 0.05, *p* < 0.01), and a significant positive correlation with total polyphenols, medium brass, amino acids, ABTS, and FRAP (*p* < 0.01). The effect of N content was opposite to that of C and P content, and the degree of influence on various nutrients and oxidative activity was mostly lower than that of CP (*p* < 0.05, *p* < 0.01). The N:P and C:P ratios were significantly positively correlated with NSC and DPPH (*p* < 0.05, *p* < 0.01), and negatively correlated with total polyphenols, total flavonoids, amino acids, ABTS, and FRAP (*p* < 0.01). However, the C:N ratio showed completely opposite behavior (*p* < 0.01). Total polyphenols, flavonoids, and amino acids were significantly positively correlated with ABTS and FRAP (*p* < 0.01), while DPPH was significantly negatively correlated (*p* < 0.05, *p* < 0.01).

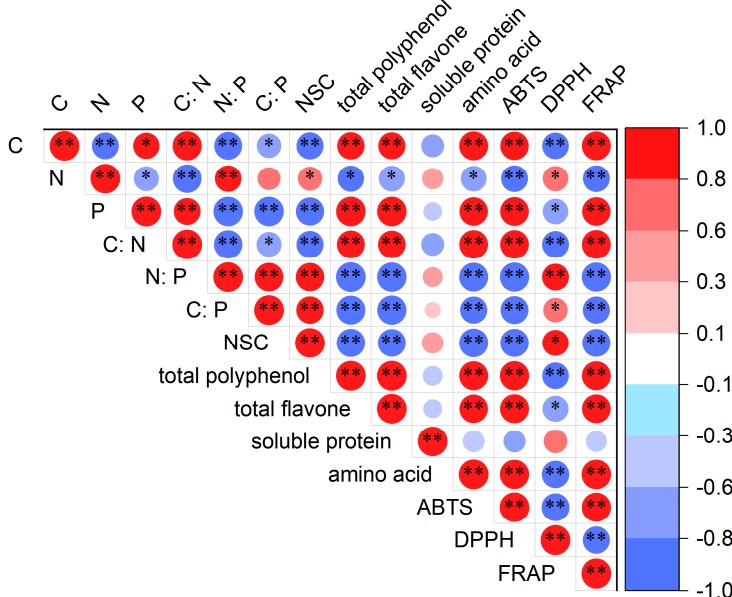

**Figure 7.** Correlation analysis between different chemical compositions and development stages in *C. chinensis* flower. Correlation matrix showing significant *p*-values (<0.05; <0.01) of different chemical compositions, where its color indicates the correlation slope (red Pearson's correlation coefficient = 1.0 and blue one = −1.0). Asterisks indicate significant differences: * *p* < 0.05, ** *p*< 0.01.

## 4. Discussion

Flowers are rich sources of phytochemistry, including amino acids, soluble protein, phenols, carbohydrates, and mineral, which havepotentialapplications in the food, cosmetics, and pharmaceutical industries [33,34]. The flower can be consumed as a bud and/or as a whole flower, or only as petals in flowering plants, which exhibits unique ingredients and nutritional value at different developmental stages [35]. *C. chinensis*, well known as an ornamentalplant in China, contained flavonoids, polyphenols, organic acids, lignans, essential oil and pigment, and showed antioxidant, tyrosinase and *α*-glucosidase inhibition activities [3]. Moreover, its flowers as medicine can treat rheumatism, muscle and bone pain, etc. [1,5]. As they are highly prized objects of beauty with commercial and industrial value, further studies are needed during flower development. The present study showed that *C. chinensis* flowers are described in detail based on the BBCH scale, and showed different morphological and physicochemical properties from mature flower buds to open flowering. These findings will help to enhance the visual quality and vase-life of flowers, and thus increasing their commercial value.

C is a skeleton element and a structural substance of life, and its content not only reflects the assimilation ability of plants, but also one of the response indicators of plants to environmental nutrient status [20]. The effectiveness of N and P affects plant growth and element balance, and the N:P ratio is related to the relative growth rate of plants [36]. Under natural conditions, it is generally believed that C does not limit plant growth, while changes in N and P are the main limiting factors for ecosystem productivity. The content and stoichiometry of elements such as C, N, and P in plants reflect their energy and nutritional status, as well as their interactions with the environment. At different developmental stages, they adjust their stoichiometry to adapt to environmental changes and meet their own metabolic needs [19]. Therefore, through N and P stoichiometric analysis can understand the nutrient utilization and limitation status of plants. Usually, a high C:N ratio and a low N:P ratio both represent nitrogen limitation, and plants have a higher nitrogen protection ability, However, high C:P and N:P ratios indicate phosphorus limitation, and plants may have a higher ability to protect phosphorus [37]. In this experiment, it is evident that the development of *C. chinensis* flowers is limited by P content, which inhibits the growth rate and biomass of *C. chinensis* flowers. The *C. chinensis* flower undergoes a significant increase in the C:P ratio and the N:P ratio during all three developmental stages. In short, during the period of *C. chinensis* flowers from bud to full opening, the limiting effect of P also increases with development and opening. Research has found that when the N:P ratio is greater than 20, the biomass can be increased by applying phosphorus [21]. So, during the development of flowers, appropriate application of P can alleviate the inhibition of the growth rate and biomass.

Sugar, as the fundamental molecule of carbon metabolism, can serve as an important energy substance throughout the entire life cycle of plants. In addition, sugar can interact with other inorganic regulatory networks as a signaling molecule, thus affecting the transformation of plant from vegetative growth to reproductive growth [38]. The proportion of different types of sugars determines the physical and chemical properties of flower buds and flowers, and the metabolic process of sugars affects the amount and form of sugars stored in flower buds and flowers [39]. When the flower bud is about to open, starch and oligosaccharides in the cells are degraded, producing a large amount of soluble sugars such as sucrose, glucose, and fructose, which can cause significant changes in the type and content of carbohydrates in the flower bud [40]. In addition to providing energy for flower development, they also provide precursors for the synthesis of biological molecules such as amino acids, nucleotides and fatty acids, so as to further synthesize other bioactive substances such as phenols and glucosinolate [39]. NSC content in this experiment also increases with flower development stage. The reason may be that a large amount of energy is consumed during the development process of flowers. *C. chinensis* flowers provide metabolic energy requirements for reproductive development through the synthesis of carbohydrates during the development process. The starch content is a pattern of sustained

growth. Soluble sugars, as carriers of plant energy and the main form of carbohydrate transport, can be directly utilized by plants [14]. The changes in soluble sugar content were gentle between stages I and II, but significantly increased during stage III, possibly due to the fact that soluble sugar is one of the important factors controlling plant flowering. During stage III, a large amount of soluble sugar is accumulated to achieve the goal of regulating flowering.

Soluble proteins are one of the factors affecting plant flowering [16]. The changes in soluble protein content in *Styrax japonicus* are the same as in this experiment, showing a "decreasing–increasing" trend [14]. The increase in the concentration of soluble protein in *Helleborus orientalis* occurs during the stage of semi opening to full opening [13]. It can be seen that soluble protein remains at a high level during the early stages of flower development, but due to the consumption of many nutrients during flower development, the content of soluble protein decreases. Subsequently, the petals absorb and store soluble protein from nutrient organs, resulting in an increase in content [14]. Amino acids and proteins are the basis of cell proliferation and Morphogenesis, play an important role in the process of flower bud differentiation, and are used to synthesize secondary metabolites, including intracellular signaling molecule, structural proteins and enzymes [9,41]. The synthesis of secondary metabolites is the reason for the stable decrease in total free amino acid content of *C. chinensis* flowers during their development period.

During the same stage of flower development, the total phenolic content exhibits three different trends: first increasing, then decreasing, continuously increasing, and continuously decreasing [8,13,17], and the changes in total flavonoid content are also similar [9,17,42]. In this experiment, both polyphenol content and flavonoid content showed a significant downward trend with the development stage of *C. chinensis* flowers. Both ABTS and FRAP show significant decreases at various developmental stages of *C. chinensis*. However, DPPH is completely the opposite. Polyphenols are the most important secondary metabolites and pigment products in plants. They include anthocyanidin, flavanols, xanthones, bicarbonate and phenolic acids [43]. Flower color presentation is closely related to the composition and accumulation of anthocyanidin and flavonol [44]. Anthocyanidin is the main component of plant petal coloring, which has strong antioxidant activity [17,45]. During the development of *C. chinensis* flower, the petal area continues to expand, and various physiological changes take place inside the petal. During the flowering process, the petal area expands faster than the speed of pigment synthesis, which leads to the decline of pigment content in the petal per unit area, The higher the water content of the petals, the lighter the color [46], resulting in varying degrees of fading of the flower color. Studies have shown that as flowers fade, the content of phenolic compounds and antioxidant activity significantly decrease [43].

## 5. Conclusions

In summary, the variations in minerals and nutrient contents, and antioxidant activity of flowers extraction in *C. chinensis* flowers were recorded, indicating that they were affected by different harvest stages. The results indicate that different developmental stages have significant effects on the C, N, and P content, the stoichiometric ratio, NSC, amino acid and polyphenol, and flavonoid contents, as well as antioxidant activity of flower extraction. The highest NSC content and DPPH activity were observed at stage III, but the highest amino acid, polyphenol, and flavonoid contents, as well ABTS and FRAP activity were observed at stage I. These findings will help to provide a broader range of options for developing the practical applications of different bioactive component at the optimal harvest stages of *C. chinensis* flowers. However, the relationship between development stages and chemical compositions is very complex in *C. chinensis* flowers, and more detailed studies will be needed to further explore its high-value utilization based on extraction and isolation, identified from aqueous extracts of *C. chinensis* flowers.

**Supplementary Materials:** The following supporting information can be downloaded at: https://www.mdpi.com/article/10.3390/horticulturae9090961/s1, Table S1 Standard curves of test parameters in this study.

**Author Contributions:** Conceptualization, G.-X.C. and S.G.; methodology, H.-Y.R. and W.-Z.Q.; software, H.-Y.R. and W.-Z.Q.; validation, H.-Y.R. and W.-Z.Q.; formal analysis, H.-Y.R. and W.-Z.Q.; investigation, H.-Y.R. and W.-Z.Q. and L.Y.; writing—original draft preparation, H.-Y.R., G.-X.C. and S.G.; writing—review and editing, H.-Y.R., W.-Z.Q., L.Y., Y.-L.Y., T.G., S.G. and G.-X.C.; visualization, S.G. and G.-X.C.; supervision, S.G. and G.-X.C. All authors have read and agreed to the published version of the manuscript.

**Funding:** This work was financially supported by the National Undergraduate Training Program for Innovation and Entrepreneurship (No. 202310626038) and the Cultivation of Scientific Research Interest Project for Undergraduate of Sichuan Agricultural University (No. 2023297, No. 2023301).

**Data Availability Statement:** The data presented in this study are available on request from the corresponding author.

**Acknowledgments:** We are grateful to all of the group members and workers for their assistance in the field experiment.

**Conflicts of Interest:** The authors declare no conflict of interest.

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
