# Peer review of "Nutrient Composition and Antioxidant Activity of Cercis chinensis Flower in Response to Different Development Stages"

_horticulturae, doi:10.3390/horticulturae9090961_

Round 1
Reviewer 1 Report
The manuscript titled “Nutrient Composition and Antioxidant Activity of Cercis chinensis Flower in Response to Different Development Stages” describes the nutritional and biofunctional characterization of Cercis chinensis flowers available in China across various developmental stages. The authors investigated a range of biofunctional parameters in the flowers, including changes in carbon, nitrogen, and phosphorous levels, soluble protein, and amino acid content, non-structural carbohydrates, as well as total phenol and total flavonoid contents at three distinct stages of flower development (Stage I - closed bud with tight clustering, displaying partial petal color at the bud tip; Stage II - expanded closed bud with scattered inflorescences, showcasing petal color across the entire surface; Stage III - fully opened flower). The authors reported fluctuations in nutrient content within the flowers based on the developmental stage. The study is intriguing, and its findings hold significance. The research design is robust and promising for processing Cercis chinensis flowers to extract valuable compounds. However, there are minor errors and grammatical mistakes that need addressing before the manuscript can be accepted for publication in Horticulturae.
Specific comments:
1. Abstract: Lines 12-13: rewrite the sentence as “However, reports on the changes of mineral elements, nutrient composition and antioxidant activity in C. chinensis flower at different development stages is rare”.
2. Abstract: Lines 16-17; the authors should mention here the duration of the study at developmental stage and intervals of sampling as the readers get a clear conception from the abstract about the study.
3. Abstract: Lines 19-20; Your sentence “The results showed that C, N and P contents showed significant decrease with the development and opening of flowers, while N content was opposite trend” is contradictory in containing N. Check please.
4. Abstract: Line 21 “The soluble protein content displayed a "decreasing-increasing" trend” the sentence should be elaborately written.
5. Abstract: Line 25; delete the word “our” from the sentence.
6. Introduction: Please add some statements mentioning the necessity of the present study before stating the purpose of the study.
7. Materials and methods: Line 94; please delete the word “site”.
8. Assay of DPPH radical-scavenging activity: Line 194; Please add the concentration of DPPH solution.
9. Assay of ABTS radical-scavenging activity: Line 207; Did the authors showed the methods of extract preparation?
10. Fig. 2 (D); The water content was significantly higher in stage 3 flower. However, the content was approximately 0.8%. Did the authors determined water after drying in same condition? If so, please explain the reason of increasing water content in stage 3 flowers.
11. Fig. 7; Please clarify the legends of the Fig. Which one indicates development stages in C. chinensis flower?
There are minor errors and grammatical mistakes that need addressing before the manuscript can be accepted.
Author Response
Author's Notes to Specific comments:
- Abstract: Lines 12-13: rewrite the sentence as “However, reports on the changes of mineral elements, nutrient composition and antioxidant activity in C. chinensis flower at different development stages is rare”.
Response:OK. Thank you for the comments, we have rewrited the sentence as suggested.
- Abstract: Lines 16-17; the authors should mention here the duration of the study at developmental stage and intervals of sampling as the readers get a clear conception from the abstract about the study.
Response:OK. Thank you for the comments, we have added relevant content in this section as suggested.
- Abstract: Lines 19-20; Your sentence “The results showed that C, N and P contents showed significant decrease with the development and opening of flowers, while N content was opposite trend” is contradictory in containing N. Check please.
Response:OK. We appreciate the reviewer's comments, and we have revised the errors in the Abstract section as suggested.
- Abstract: Line 21 “The soluble protein content displayed a "decreasing-increasing" trend” the sentence should be elaborately written.
Response:OK. We appreciate the reviewer’s comments, and have revised the sentence.
- Abstract: Line 25; delete the word “our” from the sentence.
Response:OK. We appreciate the reviewer’s comments, and we have deleted the word “our” from the sentence.
- Introduction: Please add some statements mentioning the necessity of the present study before stating the purpose of the study.
Response:OK. We appreciate the reviewer’s comments, and have added the necessity of the present study.
- Materials and methods: Line 94; please delete the word “site”.
Response:OK. We appreciate the reviewer’s comments, and have deleted the word “site”as suggested.
- Assay of DPPH radical-scavenging activity: Line 194; Please add the concentration of DPPH solution.
Response:OK. We appreciate the reviewer’s comments, and have added the concentration of DPPH solution as suggested.
- Assay of ABTS radical-scavenging activity: Line 207; Did the authors showed the methods of extract preparation?
Response:OK. We appreciate the reviewer’s comments, and have revised the section as suggested.
- Fig. 2 (D); The water content was significantly higher in stage 3 flower. However, the content was approximately 0.8%. Did the authors determined water after drying in same condition? If so, please explain the reason of increasing water content in stage 3 flowers.
Response:OK. We appreciate the reviewer’s comments. The relative water contents were measured under the same condition in this study. In certain types of flowers, increasing significantly water content is an important factor during the petal growth and development, which may play an important role in the cell expansion stage. Some reports also supported the present results, like the reports of Önder et al. (2022).
Önder Sercan,Tonguç Muhammet, ErbaÅŸ Sabri et al. Investigation of phenological, primary and secondary metabolites changes during flower developmental of Rosa damascena. Plant Physiology and Biochemistry, 2022,192.
- Fig. 7; Please clarify the legends of the Fig. Which one indicates development stages in C. chinensis flower?
Response:OK. We appreciate the reviewer’s comments. In this figure, we conducted a correlation analysis on nutrient composition and antioxidant activity at the three tested stages.
Comments on the Quality of English Language
There are minor errors and grammatical mistakes that need addressing before the manuscript can be accepted.
Reply: Ok. Thanks. The text of the whole paper has been revised, and the language was improved by a native speaker as suggested.
Reviewer 2 Report
The authors characterized nutrient levels and antioxidant activities across three developmental stages of C. chinensis via various assays. The abstract of the article can be improved as use of language is vague in some sentences. Labelling in the figures can also be improved.
In-line comments as below:
Line 14. “...development stages is rare.” – unclear what is meant by “rare”.
Line 19. “...larger variations.” – “larger” is confusing; specify what is being compared with.
Line 20. This sentence is confusing: does N content increase or decrease with development stages?
Line 21. Unclear what is meant by “decreasing - increasing” trend.
Line 92. Materials and Methods – suggest the authors include data for standard curves for each assay in supplementary materials.
Figure 2 / 3 / 4 / 5 / 6. The use of uppercase letters to represent significant differences among different development stages is confusing, as y axes of the figures are also labelled by uppercase letters. Suggest the authors choose another representation of statistical significance. Check typos in legend.
Quality of English Language can be improved. Use of words is vague in certain sentences in abstract and the main text. Suggest minor or moderate editing on grammar.
Author Response
Author's Notes to Comments and Suggestions
The authors characterized nutrient levels and antioxidant activities across three developmental stages of C. chinensis via various assays. The abstract of the article can be improved as use of language is vague in some sentences. Labelling in the figures can also be improved.
Response: Ok. Thanks. The labelling in the figures of the whole paper has been improved, and the language was improved by a native speaker as suggested.
In-line comments as below:
Line 14. “...development stages is rare.” – unclear what is meant by “rare”.
Response: OK. Thank you for the comments, we have rewrited the sentence as suggested.
Line 19. “...larger variations.” – “larger” is confusing; specify what is being compared with.
Response: OK. Thank you for the comments, we have rewrited the sentence as suggested.
Line 20. This sentence is confusing: does N content increase or decrease with development stages?
Response: OK. We appreciate the reviewer’s comments, and have revised these in the Abstract section as suggested.
Line 21. Unclear what is meant by “decreasing - increasing” trend.
Response: OK. We appreciate the reviewer’s comments, and have revised these details in the revised manuscript as suggested.
Line 92. Materials and Methods – suggest the authors include data for standard curves for each assay in supplementary materials.
Response: OK. We appreciate the reviewer’s comments, and have added standard curves in supplementary materials in the revised manuscript as suggested.
Figure 2 / 3 / 4 / 5 / 6. The use of uppercase letters to represent significant differences among different development stages is confusing, as y axes of the figures are also labelled by uppercase letters. Suggest the authors choose another representation of statistical significance. Check typos in legend.
Response: OK. We appreciate the reviewer’s comments, and have revised these figures in the revised manuscript as suggested.
Comments on the Quality of English Language
Quality of English Language can be improved. Use of words is vague in certain sentences in abstract and the main text. Suggest minor or moderate editing on grammar.
Response: Ok. Thanks. Some errors has been revised, and the language was improved by a native speaker as suggested.
Reviewer 3 Report
General comments
This manuscript evaluates the effects of different development stages on nutrient composition and antioxidant activity of Cercis chinensis flower. The study is of interest to the field of horticulture. The experimental work is in general performed well providing new information. Moreover, results are clearly commented. However, some clarifications and corrections are necessary before the manuscript can be accepted for publication.
Specific comments
Title
It do not reflect the content of the manuscript, phenol (TP) and total flavonoids (TF) are not considered
Abstract
-Lines 12-13: The authors mentioned: ..”However, the changes of mineral elements, nutrient composition and antioxidant activity in C. chinensis flower at different development stages is rare Materials and methods”..
Do the authors meant the info about these changes is scarce?
Add as conclusion at which developing stage mineral composition and antioxidant activity were the highest.
Materials and Methods
Line 183: Do not begin the sentence with a number
Statistical Analysis: It is not clear when is it used Tukey and when is it used LSD?
Results
When it is mentioned a significant decreased, it is expected that values were shown in a decreasing order. For this reason, correct the order of values in lines 255, 257, 296, 299, 315 and 319..
Revise the paragraph form lines 264 to 266.
Correct that Figure 6C is for FRAP, not for DPPH
Revise lines 386-388, trend for starch changes is contradictory
It requires minor editing
Author Response
Author's Notes to Specific comments
Title
It do not reflect the content of the manuscript, phenol (TP) and total flavonoids (TF) are not considered
Response: OK. Thank you for the comments. In fact, phenol (TP) and total flavonoids (TF) are usually considered nutrient composition in many studies, and these studies were also listed in our references.
Abstract
-Lines 12-13: The authors mentioned: ..”However, the changes of mineral elements, nutrient composition and antioxidant activity in C. chinensis flower at different development stages is rare Materials and methods”..
Do the authors meant the info about these changes is scarce?
Response: OK. Thank you for the comments, we have rewrited the sentence in the revised manuscript as suggested.
Add as conclusion at which developing stage mineral composition and antioxidant activity were the highest.
Response: OK. Thank you for the comments, and we have added the relevant content in Abstract in the revised manuscript as suggested.
Materials and Methods
Line 183: Do not begin the sentence with a number
Response: OK. Thank you for the comments, and we have rewrited the sentence in the revised manuscript as suggested.
Statistical Analysis: It is not clear when is it used Tukey and when is it used LSD?
Response: OK. Thank you for the comments, and we have revised these details in the revised manuscript as suggested.
Results
When it is mentioned a significant decreased, it is expected that values were shown in a decreasing order. For this reason, correct the order of values in lines 255, 257, 296, 299, 315 and 319..
Response: OK. Thank you for the comments, and we have revised these details in the revised manuscript as suggested.
Revise the paragraph form lines 264 to 266.
Response: We thank the reviewer and have revised the sentence in the revised manuscript as suggested.
Correct that Figure 6C is for FRAP, not for DPPH
Response: OK. Thank you for the comments, we have made revisions in the paragraph.
Revise lines 386-388, trend for starch changes is contradictory
Comments on the Quality of English Language
It requires minor editing
Reply: Ok. Thanks. The text of the whole paper has been revised, and the language was improved by a native speaker as suggested.